# Comparing trends in mid-life 'deaths of despair' in the USA, Canada and UK, 2001–2019: is the USA an anomaly?

Jennifer Beam Dowd ![ORCID],[1,2,3] Colin Angus ![ORCID],[4] Anna Zajacova,[5] Andrea M Tilstra[1,2,3]

¹Leverhulme Centre for Demographic Science, University of Oxford, Oxford, UK
²Nuffield College, University of Oxford, Oxford, UK
³Nuffield Department of Population Health, University of Oxford, Oxford, UK
⁴School of Health and Related Research, University of Sheffield, Sheffield, UK
⁵University of Western Ontario, London, Ontario, Canada

**Correspondence to**
Dr Jennifer Beam Dowd;
jennifer.dowd@demography.ox.ac.uk

## ABSTRACT

**Objectives** In recent years, 'deaths of despair' due to drugs, alcohol and suicide have contributed to rising mid-life mortality in the USA. We examine whether despair-related deaths and mid-life mortality trends are also changing in peer countries, the UK and Canada.

**Design** Descriptive analysis of population mortality rates.

**Setting** The USA, UK (and constituent nations England and Wales, Northern Ireland and Scotland) and Canada, 2001–2019.

**Participants** Full population aged 35–64 years.

**Outcome measures** We compared all-cause and 'despair'-related mortality trends at mid-life across countries using publicly available mortality data, stratified by three age groups (35–44, 45–54 and 55–64 years) and by sex. We examined trends in all-cause mortality and mortality by causes categorised as (1) suicides, (2) alcohol-specific deaths and (3) drug-related deaths. We employ several descriptive approaches to visually inspect age, period and cohort trends in these causes of death.

**Results** The USA and Scotland both saw large relative increases and high absolute levels of drug-related deaths. The rest of the UK and Canada saw relative increases but much lower absolute levels in comparison. Alcohol-specific deaths showed less consistent trends that did not track other 'despair' causes, with older groups in Scotland seeing steep declines over time. Suicide deaths trended slowly upward in most countries.

**Conclusions** In the UK, Scotland has suffered increases in drug-related mortality comparable with the USA, while Canada and other UK constituent nations did not see dramatic increases. Alcohol-specific and suicide mortalities generally follow different patterns to drug-related deaths across countries and over time, questioning the utility of a cohesive 'deaths of despair' narrative.

## INTRODUCTION

The USA has persistently lower life expectancy than its Organisation for Economic Co-operation and Development peers, including 1.7–3.2 years below the UK and Canada in 2019,[1] two countries with a shared cultural, sociopolitical and linguistic heritage. US life expectancy fell for 3 consecutive years starting in 2014, recovering slightly by 2019 prior to further declines in 2020 and 2021 during the COVID-19 pandemic.[2][3]

## STRENGTHS AND LIMITATIONS OF THIS STUDY

⇒ We directly compare trends in 'deaths of despair' in mid-life across the USA, UK and Canada.
⇒ We leverage demographic approaches to understand period versus cohort trends in these causes of death.
⇒ Coding of deaths caused by alcohol, drug overdose and suicide may vary across country and time, complicating comparisons.
⇒ Disentangling age, period and cohort effects in mortality trends is challenging.
⇒ The study is descriptive and cannot speak to the causes of trends in drug overdose, suicide and alcohol-specific deaths.

Explanations for stagnating life expectancy in the USA and divergence from peer countries have highlighted increases in 'deaths of despair' (ie, deaths due to drug and alcohol poisoning, and suicide).[4][5] Life expectancy improvements have been stalling in England and Wales[6][7] and Scotland[8] since the 2010s, raising the question of whether these US mortality dynamics reflect a broader phenomenon.

The USA, the UK and Canada have all documented rapid increases in drug-related mortality in recent years, with the USA on the leading edge with an initial increase in deaths due to prescription opioids, followed by heroin and now fentanyl.[9][10] Increases in opioid use in Canada may be related to exposure to US pharmaceutical promotional material through US media and direct marketing.[11] Within the UK, Scotland stands out with a dramatic rise in drug deaths (rising 450% since 1996) driven initially by prescription benzodiazepines and more recently by so-called 'street' benzodiazepines.[12] England and Wales have had a less extreme rise in drug-related deaths since 2012, driven by increasing numbers of opioid and cocaine-related deaths, and less noticeable increases due to benzodiazepines.[13]

Table 1  Male mortality rates (per 100 000 population) by cause and age group for England and Wales, Northern Ireland and Scotland

| Age group | England and Wales | | | Northern Ireland | | | Scotland | | |
|---|---|---|---|---|---|---|---|---|---|
| | 2001 | 2010 | 2019 | 2001 | 2010 | 2019 | 2001 | 2010 | 2019 |
| **All-cause** | | | | | | | | | |
| 35–44 | 190.7 | 168.9 | 162.8 | 192.7 | 207.7 | 190.4 | 274.9 | 244.0 | 284.2 |
| 45–54 | 461.4 | 393.9 | 371.0 | 465.7 | 443.7 | 378.1 | 649.0 | 520.1 | 529.8 |
| 55–64 | 1280.1 | 991.9 | 888.6 | 1324.3 | 1092.9 | 876.9 | 1708.3 | 1257.4 | 1076.2 |
| **Combined 'deaths of despair'** | | | | | | | | | |
| 35–44 | 44.8 | 48.0 | 58.0 | 47.6 | 67.9 | 81.9 | 82.2 | 99.5 | 157.5 |
| 45–54 | 50.5 | 58.0 | 71.1 | 54.9 | 82.6 | 85.0 | 114.9 | 109.1 | 139.2 |
| 55–64 | 46.6 | 56.0 | 66.1 | 50.0 | 79.6 | 74.5 | 127.9 | 108.3 | 98.0 |
| **Alcohol** | | | | | | | | | |
| 35–44 | 19.8 | 20.3 | 16.3 | 26.7 | 26.3 | 29.0 | 40.9 | 35.0 | 24.7 |
| 45–54 | 32.7 | 37.4 | 35.3 | 41.2 | 50.8 | 55.5 | 86.1 | 70.8 | 48.5 |
| 55–64 | 33.1 | 41.4 | 43.3 | 38.1 | 56.8 | 58.8 | 107.3 | 87.0 | 60.4 |
| **Drugs** | | | | | | | | | |
| 35–44 | 9.8 | 12.8 | 22.8 | 1.6 | 10.6 | 23.6 | 15.1 | 36.0 | 102.4 |
| 45–54 | 4.6 | 6.1 | 16.6 | 1.8 | 6.3 | 6.2 | 5.5 | 16.6 | 63.6 |
| 55–64 | 2.3 | 2.5 | 6.5 | 2.0 | 4.1 | 0.8 | 2.3 | 6.6 | 15.6 |
| **Suicide** | | | | | | | | | |
| 35–44 | 15.2 | 14.9 | 18.9 | 19.4 | 31.0 | 29.3 | 26.2 | 28.5 | 30.4 |
| 45–54 | 13.2 | 14.5 | 19.1 | 11.9 | 25.5 | 23.4 | 23.2 | 21.7 | 27.1 |
| 55–64 | 11.1 | 12.1 | 16.3 | 9.9 | 18.7 | 15.0 | 18.3 | 14.7 | 22.0 |

Besides changes in the supply and marketing of prescription and illicit drugs, increases in drug mortality may reflect demand-side factors related to the loss of economic opportunity, particularly among socioeconomically disadvantaged groups.[14 15] In Scotland, cohort trends in drug-related deaths and suicide were evident particularly in men from socioeconomically deprived areas coming of age in the 1980s.[16 17] In the USA and the UK, lack of economic opportunity in middle-aged cohorts may contribute to poorer mental health and subsequent unhealthy coping behaviours.[18 19] In Canada, socioeconomic inequalities in health measured by educational attainment and household income have increased over time, but less is known about whether this has translated into despair-related causes of death.[20]

This paper compares recent trends in mid-life 'deaths of despair' and its component causes in the USA with those in the UK and Canada from 2000 to 2019. The UK was chosen as a comparison country also witnessing stalls in life expectancy. Thus far, whether UK patterns of mid-life mortality and despair related are following the USA is not known. Canada shares close economic and geographical ties to the USA and has seen spillover effects of changing drug supply and the opioid epidemic.[21]

## Data

Publicly available all-cause and cause-specific mortality by age and sex were analysed for all countries. Because data are collected by separate statistical offices for the nations of the UK, and because of historically divergent mortality trends for Scotland, we estimate trends separately for UK nations, except for England and Wales whose data are released together. For England and Wales, we analysed the 21st Century Mortality dataset from 2001 to 2019,[22] a publicly available resource that includes all registered deaths by cause as well as midyear population counts by 5-year age groups. Comparable data for Scotland are available from National Records of Scotland's Vital Events Reference Tables[23] and for Northern Ireland from the Registrar General's Annual Reports.[24] For Canada, death counts by cause of death were accessed from publicly available Statistics Canada Table 13-10-0392-01.[25] These records include all registered deaths in Canada each year from 2000 to 2019 by sex and 5-year age group. US mortality data by year and single year of age come from CDC Wonder.[26] Population counts by age and sex for all countries were obtained from the Human Mortality Database. Results are fully reproducible, and all metadata and code used in analysis can be accessed here.

## METHODS

We focus on mid-life mortality where increases have been documented in the USA, specifically age groups 35–44, 45–54 and 55–64 years. We examine trends in mortality

**Table 2** Male mortality rates (per 100 000 population) by cause and age group for Canada and the USA

| Age group | Canada | | | USA | | |
|---|---|---|---|---|---|---|
| | 2001 | 2010 | 2019 | 2001 | 2010 | 2019 |
| All-cause | | | | | | |
| 35–44 | 183.6 | 164.1 | 161.7 | 259.2 | 212.2 | 257.3 |
| 45–54 | 424.3 | 361.3 | 334.6 | 551.3 | 508.9 | 484.9 |
| 55–64 | 1155.9 | 916.9 | 826.3 | 1226.1 | 1082.7 | 1106.7 |
| Combined 'deaths of despair' | | | | | | |
| 35–44 | 44.4 | 46.3 | 50.1 | 58.2 | 61.0 | 101.6 |
| 45–54 | 53.7 | 58.2 | 62.6 | 77.2 | 94.1 | 115.5 |
| 55–64 | 67.9 | 68.3 | 75.5 | 68.7 | 93.0 | 134.8 |
| Alcohol | | | | | | |
| 35–44 | 8.2 | 8.5 | 8.4 | 14.9 | 12.4 | 16.9 |
| 45–54 | 22.7 | 22.1 | 21.0 | 36.9 | 36.8 | 37.1 |
| 55–64 | 42.5 | 38.5 | 40.2 | 42.9 | 50.3 | 63.2 |
| Drugs | | | | | | |
| 35–44 | 9.7 | 13.4 | 21.1 | 20.1 | 23.6 | 56.6 |
| 45–54 | 6.1 | 11.7 | 19.6 | 16.7 | 26.7 | 49.2 |
| 55–64 | 3.2 | 7.1 | 13.1 | 4.6 | 15.1 | 41.1 |
| Suicide | | | | | | |
| 35–44 | 26.6 | 24.4 | 20.6 | 23.3 | 25.1 | 28.1 |
| 45–54 | 24.9 | 24.4 | 22.0 | 23.7 | 30.6 | 29.3 |
| 55–64 | 22.3 | 22.7 | 22.1 | 21.2 | 27.6 | 30.5 |

by causes categorised as (1) suicides (excluding undetermined intent), (2) alcohol-specific deaths, (3) drug-related deaths and a combined (4) 'deaths of despair' category summing all three causes, as well as all-cause mortality (The 10th revision of the International Statistical Classification of Diseases and Related Health Problems (ICD-10) codes are available in online supplemental table 1). In order to mitigate the impact of random variation in the cause-specific and age-specific mortality rates, we used a one-dimensional P-spline approach for each country, cause, sex and calendar year to smooth the observed death rates over age.[27] The resulting single year of age-specific mortality rates was then age standardised within each of the three broad age groups using the European Standard Population.[28] We visually inspected cohort versus period trends using plots of the 3-year rolling averages within age groups over time, supplemented with Lexis surfaces.

### Patient and public involvement
None.

### RESULTS
Full results for all age, sex, country and cause groups for all calendar years are shown in online supplemental tables 2A–6B, with summary data for 2001, 2010 and 2019 presented in tables 1–4. All-cause and cause-specific mortality trends for the combined age range of 35–64

years of age are shown in online supplemental figures 1 and 2A–C.

Figure 1 shows all-cause mortality rates from 2001 to 2019 by age and sex group. Observed mortality patterns differ across nations and by sex. In Canada and England and Wales, mortality rates are low and decline consistently for most age–sex combinations, except for those 35–44 years old for which mortality trends are flat or increasing. Upticks in mortality were more noticeable in Scotland and the USA in those 35–44 years old, with steeper increases for men compared with women. In the USA, all-cause mortality among men 35–44 years reached a low point of 211.8/100 000 in 2012 before rising steadily to 257.3/100 000 in 2019 (online supplemental table 6A). Men aged 35–44 years in Scotland followed a similar trend, with slightly higher absolute levels of mortality compared with US men. This youngest age group also showed signs of stalling improvements or slight increases in mortality in Canada, England and Wales, and Northern Ireland. Noticeable increases in overall mortality were also seen in Scotland for men and women aged 45–64 years, but not in the 55–64 years old age group. The USA was notable for its stagnation or increases in mortality across almost all age/sex groups.

Figure 2 shows trends in despair-related deaths over the same period. The USA and Scotland again stand out for substantially higher and largely increasing rates across the period. Men aged 35–44 years show some of

**Table 3** Female mortality rates (per 100 000 population) by cause and age group for England and Wales, Northern Ireland and Scotland

| Age group | England and Wales | | | Northern Ireland | | | Scotland | | |
|---|---|---|---|---|---|---|---|---|---|
| | 2001 | 2010 | 2019 | 2001 | 2010 | 2019 | 2001 | 2010 | 2019 |
| All-cause | | | | | | | | | |
| 35–44 | 117.5 | 104.8 | 98.7 | 119.4 | 122.3 | 115.0 | 149.2 | 133.4 | 153.4 |
| 45–54 | 302.5 | 260.4 | 243.0 | 310.2 | 292.7 | 274.0 | 387.9 | 328.6 | 341.1 |
| 55–64 | 802.5 | 641.2 | 587.8 | 809.9 | 704.9 | 639.7 | 1014.3 | 835.8 | 759.9 |
| Combined 'deaths of despair' | | | | | | | | | |
| 35–44 | 16.8 | 18.3 | 23.2 | 20.1 | 26.3 | 32.3 | 35.6 | 37.4 | 61.7 |
| 45–54 | 23.0 | 24.6 | 30.5 | 29.4 | 39.6 | 43.3 | 48.4 | 46.4 | 64.7 |
| 55–64 | 23.7 | 25.4 | 29.4 | 24.6 | 36.3 | 42.5 | 49.9 | 44.9 | 53.3 |
| Alcohol | | | | | | | | | |
| 35–44 | 9.6 | 10.2 | 9.7 | 15.0 | 13.9 | 17.1 | 19.9 | 17.9 | 12.8 |
| 45–54 | 16.4 | 17.3 | 18.2 | 22.9 | 25.9 | 29.9 | 36.7 | 32.2 | 28.4 |
| 55–64 | 17.7 | 19.6 | 20.5 | 20.8 | 26.1 | 32.7 | 41.3 | 35.6 | 37.5 |
| Drugs | | | | | | | | | |
| 35–44 | 3.5 | 4.3 | 8.4 | 1.2 | 4.4 | 8.2 | 4.8 | 10.5 | 40.2 |
| 45–54 | 2.5 | 3.3 | 7.1 | 2.4 | 4.2 | 6.4 | 2.8 | 7.5 | 26.1 |
| 55–64 | 1.9 | 2.1 | 4.2 | 0.2 | 3.1 | 4.1 | 2.0 | 4.7 | 9.0 |
| Suicide | | | | | | | | | |
| 35–44 | 3.8 | 3.8 | 5.2 | 3.9 | 7.9 | 7.0 | 10.9 | 8.9 | 8.7 |
| 45–54 | 4.2 | 4.0 | 5.3 | 4.1 | 9.5 | 7.0 | 9.0 | 6.7 | 10.2 |
| 55–64 | 4.1 | 3.7 | 4.7 | 3.6 | 7.1 | 5.8 | 6.6 | 4.6 | 6.8 |

the worst trends among analysed countries, with rates in the USA and Scotland increasing almost twofold over the period. While the absolute increases for men in England and Wales and Northern Ireland are much smaller, they reflect large relative increases (for example, a 41% increase in despair-related deaths for men aged 45–54 years over this period in England and Wales). Canada is notable for its low levels and relatively small increases in deaths of despair, especially for women for whom rates hardly increase. While women in most nations saw only small increases in deaths of despair, the USA stood out with a trend of high and rising mortality for both men and women in all age groups.

Figure 2A–C breaks down these 'despair' deaths into individual causes, where we see the salience of increasing drug-related mortality in the USA and Scotland. The magnitude of the increases for men aged 35–44 years in both countries was staggering (an almost sixfold increase in Scotland from 15.1/100 000 to 102.3/100 000). In the USA, these increases are seen across all age groups, while in Scotland, the increases are significantly less pronounced for those 55–64 years old. Although at lower absolute levels, drug-related mortality also increased in Canada, England and Wales, and Northern Ireland over the period, especially among younger men (for example, an increase of 118% from 9.1/100 000 to 21.1/100 000 for Canadian men aged 35–44 years).

Suicide mortality saw more year-to-year variability, with evidence of small increases among men in England and Wales, Scotland and the USA. The magnitude of these changes is very small compared with changes in drug-related deaths. Suicide rates were higher for men in all nations. Canada consistently had the lowest levels of suicide mortality across all age/sex groups. The USA generally had the highest levels of suicide deaths, except for men aged 35–44 years where Scotland and Northern Ireland had higher levels. Alcohol-specific mortality across most nations was relatively stable, but with sharp declines over time in Scotland from previously very high levels, especially at the older ages. In contrast, increases in alcohol-specific mortality in the USA were seen in ages 55–64 years, especially for men. Taken together, these results suggest that drug-related mortality is the primary cause of increases in overall 'despair'-related mortality over this period, with small but perceptible increases in England and Wales, Canada and Northern Ireland, and very pronounced increases in the USA and Scotland.

Online supplemental figure 3A–C shows an alternative view of the same data to examine period versus cohort trends. Here we look for 'non-parallelism' in the age-specific mortality curves over time within each country.[29] If the age-specific curves are parallel to each other, this likely reflects the salience of period influences. Non-parallel age-specific trends suggest more importance for

**Table 4** Female mortality rates (per 100 000 population) by cause and age group for Canada and the USA

| Age group | Canada | | | USA | | |
|---|---|---|---|---|---|---|
| | 2001 | 2010 | 2019 | 2001 | 2010 | 2019 |
| All-cause | | | | | | |
| 35–44 | 112.1 | 100.2 | 94.6 | 147.6 | 128.1 | 142.8 |
| 45–54 | 268.6 | 235.9 | 212.6 | 319.8 | 311.7 | 295.1 |
| 55–64 | 685.9 | 586.7 | 529.9 | 773.4 | 647.8 | 667.8 |
| Combined 'deaths of despair' | | | | | | |
| 35–44 | 15.9 | 19.1 | 19.0 | 22.0 | 27.7 | 40.5 |
| 45–54 | 20.0 | 24.3 | 24.1 | 26.3 | 42.8 | 51.2 |
| 55–64 | 24.1 | 26.7 | 28.9 | 24.1 | 35.1 | 53.8 |
| Alcohol | | | | | | |
| 35–44 | 4.0 | 4.2 | 4.3 | 6.4 | 5.5 | 8.6 |
| 45–54 | 8.6 | 8.5 | 9.5 | 12.1 | 14.6 | 17.7 |
| 55–64 | 14.7 | 13.2 | 16.4 | 15.8 | 17.3 | 26.7 |
| Drugs | | | | | | |
| 35–44 | 4.1 | 7.2 | 9.0 | 9.2 | 14.7 | 23.6 |
| 45–54 | 3.4 | 7.3 | 8.2 | 7.0 | 19.0 | 23.2 |
| 55–64 | 2.5 | 5.7 | 5.8 | 2.6 | 9.9 | 18.2 |
| Suicide | | | | | | |
| 35–44 | 7.8 | 7.6 | 5.7 | 6.5 | 7.4 | 8.2 |
| 45–54 | 8.0 | 8.4 | 6.5 | 7.2 | 9.2 | 10.3 |
| 55–64 | 6.9 | 7.8 | 6.7 | 5.7 | 7.9 | 8.9 |

cohort effects (for a detailed discussion and simulations of these dynamics, see 30, pages 204–217). For overall deaths of despair, age-specific trends over time are mostly parallel for Canada, England and Wales, and Northern Ireland, suggesting period-based dynamics affecting all cohorts. The USA and Scotland show some deviations from parallelism, with the youngest age group showing sharper increases than the older groups in Scotland,

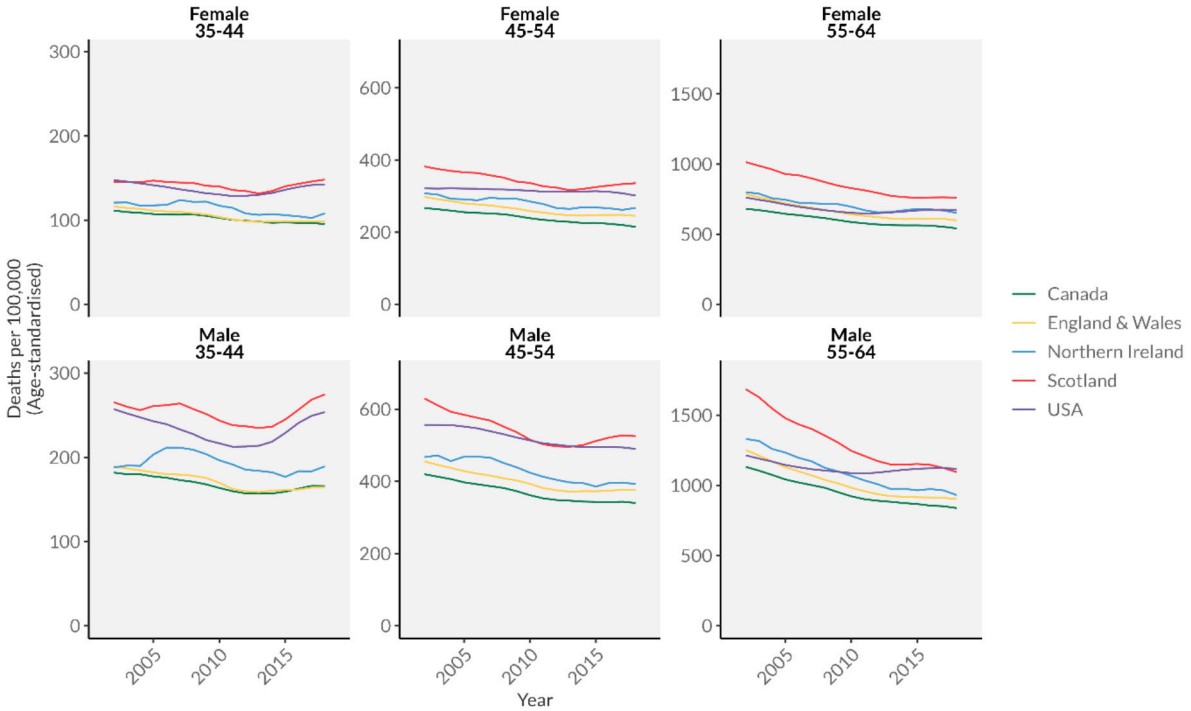

**Figure 1** All-cause mortality by age and sex in Canada, the UK and the USA, 2001–2019.

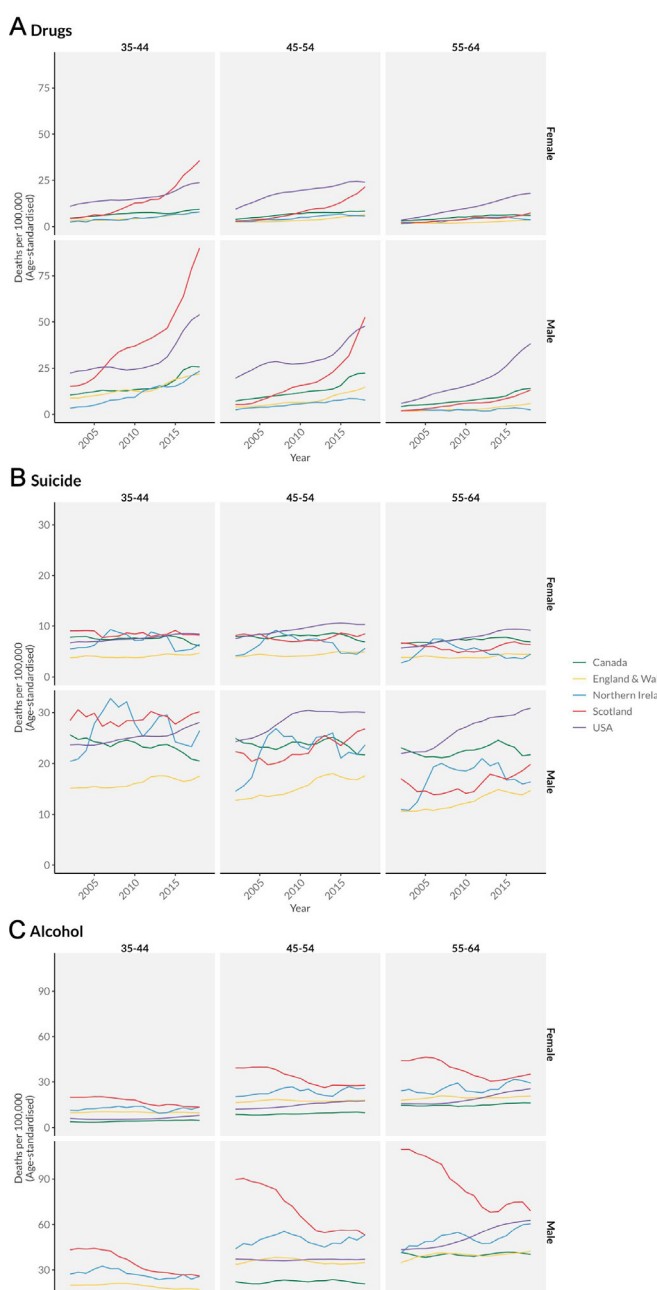

**Figure 2** Combined 'deaths of despair' mortality (drug, suicide and alcohol) by age and sex in Canada, the UK and the USA, 2001–2019. (A) Drug-related mortality by age and sex in Canada, the UK and the USA, 2001–2019. (B) Suicide mortality by age and sex in Canada, the UK and the USA, 2001–2019. (C) Alcohol-specific mortality by age and sex in Canada, the UK and the USA, 2001–2019.

and those 55–64 years old in the USA increasing more steeply (and earlier) than younger groups. For drug-related deaths, parallel trends are largely suggestive of period-based effects for Canada, England and Wales, and the USA. Non-parallelism in Northern Ireland and Scotland suggests younger born cohorts have been hit harder by recent increases in drug deaths than the oldest cohort. For suicide, most trends are parallel, suggesting

period fluctuations, except for the USA where the oldest age group has increased more steeply. Alcohol-specific mortality mostly follows period-based parallel trends, except for the USA where again the oldest age group has increased over time. Cohort-driven declines in alcohol-related mortality were seen in Scotland, with the older groups declining steeply from high levels compared with lower and flatter trends in the youngest age group. Online supplemental figure 4A,B shows alternative visualisations for cohort patterns in mortality using Lexis surfaces. Overall, these visualisations highlight some limited cohort trends but also very strong period effects of increased drug deaths that affected most age groups and nations. In the USA, increases in drug-related deaths affected all age groups over time, but sharper increases in alcohol-specific deaths and suicide were only seen in the oldest age group (earliest born cohort). This oldest group largely reflects the baby boom cohort (born 1946–1964) passing this 55–64 age group over the observed period.

Online supplemental figure 5A–C shows trends in the proportion all deaths due to the individual despair-related causes over time, with drug-related mortality standing out as an increasing proportion of all-cause mortality. From table 1, we can see that the all-cause mortality rate increased by 45/100 000 for US men aged 35–44 years old from 2010 to 2019. Of this increase, 73% could be accounted for by an increase in drug-related mortality, and 90% was accounted for by increases in all three despair-related causes. For men aged 55–64 years old, all-cause mortality increased by 23.9/100 000 over the same period, which was more than fully accounted for by an increase in drug-related mortality of 26/100 000 from 2010 to 2019. Thus, in many age/sex groups, all-cause mortality would have declined or plateaued rather than increased in the USA if drug mortality had stayed constant.

## DISCUSSION

We describe large relative increases and high absolute levels of drug-related deaths in both the USA and Scotland from 2001 to 2019, with lower levels and smaller increases in Canada and the rest of the UK. Alcohol-specific deaths showed less consistent trends that did not track other 'despair' causes, with older groups in Scotland seeing steep declines over time. Suicide deaths trended slowly upward in most countries. We generally found more evidence for period rather than cohort effects for these trends, apart from baby boomers in the USA, consistent with previous evidence of their cohort mortality disadvantage.[31]

Increases in mid-life mortality in the last two decades in the USA set off a robust debate about the underlying causes of this concerning trend, with a strong focus on 'deaths of despair'. One narrative emphasises the role of cumulative disadvantage triggered by worsening labour market opportunities for white Americans with low levels of education, leading to psychological distress

and unhealthy behavioural coping mechanisms.[5 32 33] A second hypothesis emphasises the strong period trend in drug-related mortality due to changes in the availability of prescription and illicit opioids, including heroin and synthetic opioids like fentanyl.[34 35] While the first hypothesis implies that 'despair' deaths share a fundamental cause and typically move together, the latter argues that the trends in suicide and alcohol-specific mortality are distinct and do not fit well within the overall deaths of despair narrative.[36 37]

Our study extends this work by comparing mid-life mortality trends in the USA with Canada and the UK, two countries with shared sociocultural histories with the USA but different economic, political and healthcare contexts. The UK has also experienced recent slowdowns in mortality improvements, raising the question of whether the USA is the leading edge of more general mortality trends. Overall, we find evidence that the levelling off and increases in mid-life mortality seen in the USA may be emerging more broadly, but the magnitude of these changes in other countries is muted. Scotland is the exception, with upward trends in drug-related deaths surpassing the high levels in the USA and contributing to increases in overall mortality trends for men aged 35–54 years. While drug-related deaths have increased on a relative scale in the other UK nations and Canada, deaths from these causes are at much lower absolute levels than the USA and Scotland. There is little evidence of significant trends in suicide or alcohol-specific deaths contributing to changes in all-cause mortality.

Our results suggest that increases in drug-related mortality are not necessarily unique to the USA, with smaller increases also seen in Canada, England and Wales, and Northern Ireland, particularly among younger men. Scotland stands out as trending even higher than the USA in drug-related mortality at younger ages, while the USA saw parallel increases in drug mortality across all age ranges, suggesting a strong period effect. We found some evidence of cohort effects in alcohol-specific and suicide mortality in the USA, with the oldest cohort trending up relative to younger cohorts. Suicide mortality rates were generally stable across countries for women, with slight increases for men and women in the USA. Overall, with the exception of the oldest born (ie, baby boom) male cohort in the USA, we find that individual 'deaths of despair' do not follow similar age and cohort patterns and thus may not reflect the same fundamental causes.

The fact that drug-related deaths have increased in other countries but not to the degree of the USA is consistent with differences in prescribing practices and healthcare contexts. In the UK, there is no direct-to-consumer marketing by pharmaceutical companies, and the primary care system likely mitigates patients receiving prescriptions from multiple sources.[38] Nonetheless, prescribing for non-cancer pain did increase dramatically since the late 1990s when there were consensus statements supporting the use of opioids for the treatment of chronic pain.[39 40] Despite this increase in overall opioid prescribing, a review of the UK Clinical Practice Research Datalink did not find an increase in the rate of diagnosed opioid use disorders in the UK between 2008 and 2012.[41] While deaths from fentanyl in the UK have grown, they are still small relative to deaths from heroin or morphine, suggesting the context for drug prescribing and misuse may be different in the UK.[42] The dramatic rise in drug deaths in Scotland, with rates far outstripping those elsewhere in Europe (https://www.nrscotland.gov.uk/files//statistics/drug-related-deaths/20/drug-related-deaths-20-pub.pdf), has arisen through a somewhat different mechanism to the USA's initial prescription opioid-driven crises. A combination of economic, social and political factors experienced by those living in deprived areas in the 1980s led to a rise in drug-related death rates in the 1990s,[16 17] driven by opioids such as heroin, often taken in combination with prescription benzodiazepines such as diazepam.[43] Although steps were taken to reduce the prescribing of benzodiazepines in the late 2000s, these were swiftly replaced by a range of 'synthetic benzodiazepines' or 'street benzos' such as etizolam, which became widely and cheaply available through illicit markets.[12] The explosion in the use of street benzos has played a huge role in the rapid rise in Scottish drug deaths since 2012 when street benzos were implicated in 3% of all drug deaths compared with 66% in 2020, with concurrent use with opioids in particular the key factor—less than 1% of drug deaths in 2020 involved street benzos alone (https://www.nrscotland.gov.uk/statistics-and-data/statistics/statistics-by-theme/vital-events/deaths/drug-related-deaths-in-scotland/2020). In the UK overall, austerity measures after the 2008 financial crisis have been suggested as one cause of stalling life expectancy.[44 45] While many have focused on cuts in health and social care affecting older age mortality,[46] some evidence has linked austerity to an increase in drug deaths affecting younger and middle ages.[47 48] While concern over the opioid epidemic and fentanyl in Canada is rising,[49] the absence of overall increases in mortality thus far has minimised discussion of a similar epidemic of 'deaths of despair'.

### Strengths and limitations

Our study has several strengths and limitations. We use a demographic perspective to compare period and cohort trends in 'despair'-related deaths in the USA, UK and Canada. Disentangling age, period and cohort effects in mortality is a persistent challenge, and we take a transparent visualisation approach to ease interpretation. When examining external causes of death such as those due to alcohol, drug overdose and suicide, there are differences in coding practices across country and time that could impact results. The overlapping nature of these causes also means there will be limitations to any set of definitions. We largely follow the original 'despair' categorisation of Case and Deaton[5] and have previously demonstrated that the use of alternative categorisation is unlikely to substantively change our conclusions.[50] Our suicide category excludes deaths from undetermined

intent, which differs from some studies[19] and should be considered when doing international comparisons. Additionally, our study is descriptive and cannot speak to the causes of trends in drug overdose, suicide, and alcohol-specific deaths or related policy solutions. Finally, we examine trends only through 2019 and do not consider the specific dynamics of trends in despair-related mortality during the COVID-19 pandemic. Recent work showed drug deaths rising considerably in the USA during the pandemic across all ages, which was not seen in England or Scotland.[50] Concerns about rising suicide rates during the pandemic seem unfounded, but alcohol-related deaths increased in both the USA and UK.[50] The impact of the pandemic as a period and cohort shock to risk behaviours and causes of deaths will be an important area of future research.

## Conclusions

'Deaths of despair' due to alcohol, drug overdoses and suicide have been implicated in rising mid-life mortality in the USA, but how these trends compare with peer countries is less well-known. We find that the USA is unique in its large relative and absolute increases in drug-related mortality for all mid-life age/sex groups, but that Scotland has seen a dramatic upward trend in drug deaths that now exceed the USA for younger age groups. The huge increase in drug mortality across all ages in the USA from 2001 to 2019 is most consistent with period explanations, while alcohol and suicide upward trends in the USA are more focused in the oldest cohort. Overall, the trends in individual causes of despair deaths are not consistent across countries or age groups, questioning the value of a cohesive 'deaths of despair' narrative. This is consistent with other work finding different spatiotemporal patterns of these causes within the USA.[51] Future research should consider the determinants of these specific causes across place and time without assuming they represent an underlying phenomenon with the same underlying drivers.[52]

**Contributors** The authors confirm contribution to the paper as follows: study conception and design—JBD. Data collection—CA, JBD and AZ. Analysis and interpretation of results—CA and JBD. Draft manuscript preparation—JBD. All authors critically reviewed the results and approved the final version of the manuscript. JBD is the guarantor who accepts full responsibility for the work and/or the conduct of the study, had access to the data, and controlled the decision to publish.

**Funding** The authors acknowledge funding support from the European Research Council (ERC-2021-CoG-101002587) and the Leverhulme Trust (Large Centre Grant).

**Disclaimer** The funders played no direct role in this research.

**Competing interests** None declared.

**Patient and public involvement** Patients and/or the public were not involved in the design, or conduct, or reporting, or dissemination plans of this research.

**Patient consent for publication** Not required.

**Ethics approval** The study does not involve human subjects, as it uses publicly available death counts from national vital statistics and no individual-level data.

**Provenance and peer review** Not commissioned; externally peer reviewed.

**Data availability statement** Data are available in a public, open access repository. All aggregated metadata and code can be accessed at: https://github.com/VictimOfMaths/DeathsOfDespair/tree/master/Paper.

**ORCID iDs**
Jennifer Beam Dowd http://orcid.org/0000-0003-2006-5656
Colin Angus http://orcid.org/0000-0003-0529-4135

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
