## [Reviewer comments · BMJ Open]

ARTICLE DETAILS

TITLE (PROVISIONAL)	Comparing trends in Mid-life "Deaths of Despair" in the US, Canada, and UK, 2001-2019: Is the US an Anomaly?
AUTHORS	Dowd, Jennifer; Angus, Colin; Zajacova, Anna; Tilstra, Andrea

VERSION 1 – REVIEW

REVIEWER	Jasilionis, D Max-Planck-Institut fur Demografische Forschung
REVIEW RETURNED	04-Jan-2023

GENERAL COMMENTS	The manuscript compares trends in “deaths of despair” in mid-life across the US, UK, and Canada. This is an important and new piece of evidence complementing recent literature on stalling life expectancy in the US and some other high-income countries. The data and insights presented in the paper are likely to be very interesting for international readership. The manuscript is very well written and structured. It is also important that the authors acknowledge the major limitations related to cause-specific data and do not claim any causal relationships. I have a couple of comments related to the scope and interpretations. First, although the impact of Covid-19 is a separate story, it is important at least briefly discuss changes in all-cause mortality beyond 2019. This can be done by updating Figure 1 with more recent data. Some insights or evidence from other studies could be added discussing possible impacts of the Covid-19 pandemic on “deaths of despair”. Second, the conclusion stating that “...the trends in individual causes of despair deaths are not consistent across countries or age-groups, casting doubt on the underlying unity of a “deaths of despair” narrative” deserves more justification in the discussion section.
--

REVIEWER	Martinson, Melissa University of Washington
REVIEW RETURNED	30-Jan-2023

GENERAL COMMENTS	This manuscript provides an important update and additional context to the Case and Deaton framework on increasing “deaths of despair” in the United States. The analysis of mid-life deaths in the US, UK, and Canada by the different components of deaths of despair over an almost 20 year period ending just before the Covid-19 pandemic is an important contribution as comparative mortality research, and will be important to our knowledge as we reckon with the impact of the pandemic and the examination of changes in these causes of death during its different stages. The analysis includes supplemental findings that show strong period effects and limited cohort effects as detailed in the results. Importantly, it provides analysis that shows
--

	that the “deaths of despair” framing of changes in mortality by suicide, drug-related deaths, and alcohol specific deaths may not be appropriate as patterns vary by these different causes in the US (as well as CA and UK). It concludes with a discussion of policy contexts that may be important to these descriptive findings. Methods:  1. Just for clarity the mortality causes in category #4 (combined deaths of despair) is inclusive of the 3 previous causes: either suicide, alcohol-specific, or drug-related? And all-cause is a 5th category. This seems to be the case based on the tables, but it will help the reader to clarify here. 2. Appreciated the ICD-10 codes are in the supplemental table. I skipped over this the first read as it's in parentheses. I wondered about any differences in causes of death coding based on the “strengths and limitations” bullets, but didn't see it addressed again. Could this be described in the paper (methods or limitations)? Is there a concern that a systematic difference in coding of deaths could influence the results? Results:  3. This may be beyond the scope of this manuscript, but with the inclusion of all-cause mortality, I was interested to see a bit more analysis of how the different causes of “deaths of despair” might be playing out as a proportion of mortality trends in the countries examined. A short section highlighting this might be a nice addition to the paper. It is described a bit in the Discussion section, so a short inclusion in results would tie this together.
--	---

REVIEWER	Walsh, David Glasgow Centre for Population Health
REVIEW RETURNED	31-Mar-2023

GENERAL COMMENTS	Thanks for the opportunity to review this study of trends in deaths from suicide, alcohol-related causes and drug-related causes in the USA, Canada and the four UK nations. The analyses seem detailed and comprehensive, reflecting a lot of work, and the paper is therefore of interest. My main query relates to the choice of countries which isn't really justified to any great degree – other than saying they have shared cultural and linguistic characteristics. On that basis, why not Australia and/or New Zealand? And of course there is a strong argument that, culturally, the USA is quite a different entity and the UK is more similar to other European countries (despite Brexit!). Was it a data-driven choice? It seems really important to clarify and justify the selection of countries and I'm not sure the paper, in its current form, really does that? The ICD code based definitions are also important. There has been a lot of work done in recent years in producing agreed, comparable, definitions of these causes of death. The authors' definitions seem different to others I have seen. Importantly, for suicide I don't think they include deaths from undetermined intent and for international comparisons, I think they should. More broadly, how do these definitions compare with studies employing different definitions – what (if any) are the implications? There is a considerable body of evidence (international and UK based) which have attributed changes in mortality rates since the early 2010s to austerity measures which (in the UK especially) have
--

	impacted on the poorest and most vulnerable. This includes various studies of the association between austerity in England & Wales (and also Scotland) and increased drug-related deaths. It seems a huge omission for this not to be alluded to in the Discussion (especially as the impacts have been observed for around half the period you are looking at). The Discussion needs amended: there is no strengths & limitations section, no summary of main findings at the start, no policy implications. There are standard sub-headings that can be used for this type of paper that people would expect to see. Some more minor points:  • Why choose 2001 as the starting point – is that data driven? Longer terms trends back to the 1980s and 1990s highlight a lot of important issues. • You analyse by quite small age groups – that’s fine, but an overall ‘premature mortality’ (<65 years) might also be important/helpful to look at • Given one of your conclusions is that “deaths of despair” isn’t actually a meaningful grouping (as trends are different for the three different causes included), is there an argument to keep them separate in the analyses (and not report them as a combined measure)? • In the abstract it would be helpful to define ‘deaths of despair’ earlier • Supreme-minor point: before a copy-editor picks up on it, your data sources should be fully referenced, not just as footnotes. • Another supreme minor point: reference 19 is at the end of a sentence discussing Canada and so isn’t appropriate
--	--

VERSION 1 – AUTHOR RESPONSE

Reviewer: 1

Dr. D Jasilionis, Max-Planck-Institut für Demografische Forschung

Comments to the Author:

The manuscript compares trends in “deaths of despair” in mid-life across the US, UK, and Canada. This is an important and new piece of evidence complementing recent literature on stalling life expectancy in the US and some other high-income countries. The data and insights presented in the paper are likely to be very interesting for international readership. The manuscript is very well written and structured. It is also important that the authors acknowledge the major limitations related to cause-specific data and do not claim any causal relationships.

I have a couple of comments related to the scope and interpretations. First, although the impact of Covid-19 is a separate story, it is important at least briefly discuss changes in all-cause mortality beyond 2019. This can be done by updating Figure 1 with more recent data. Some insights or evidence from other studies could be added discussing possible impacts of the Covid-19 pandemic on “deaths of despair”. Second, the conclusion stating that “...the trends in individual causes of despair deaths are not consistent across countries or age-groups, casting doubt on the underlying unity of a “deaths of despair” narrative” deserves more justification in the discussion section.

Many thanks to the reviewer for their kind remarks.

We agree that trends in deaths of despair since 2020 have been notable in some countries. Given the complexity of the interplay between the COVID-19 pandemic and deaths related to

suicide, drugs, and alcohol, and delays in release of cause of death data for some countries (specifically Canada) we feel it is beyond the scope of this paper. While our scientific focus remains on the pre-pandemic period, we have now mentioned and provided citations for evidence on trends in these causes during the pandemic.

Reviewer: 2

Dr. Melissa Martinson, University of Washington

Comments to the Author:

This manuscript provides an important update and additional context to the Case and Deaton framework on increasing “deaths of despair” in the United States. The analysis of mid-life deaths in the US, UK, and Canada by the different components of deaths of despair over an almost 20 year period ending just before the Covid-19 pandemic is an important contribution as comparative mortality research, and will be important to our knowledge as we reckon with the impact of the pandemic and the examination of changes in these causes of death during its different stages. The analysis includes supplemental findings that show strong period effects and limited cohort effects as detailed in the results. Importantly, it provides analysis that shows that the “deaths of despair” framing of changes in mortality by suicide, drug-related deaths, and alcohol specific deaths may not be appropriate as patterns vary by these different causes in the US (as well as CA and UK). It concludes with a discussion of policy contexts that may be important to these descriptive findings.

Methods:

1. Just for clarity the mortality causes in category #4 (combined deaths of despair) is inclusive of the 3 previous causes: either suicide, alcohol-specific, or drug-related? And all-cause is a 5th category. This seems to be the case based on the tables, but it will help the reader to clarify here.

Many thanks, that is correct, and we have clarified this in the text and Figure labels.

2. Appreciated the ICD-10 codes are in the supplemental table. I skipped over this the first read as it's in parentheses. I wondered about any differences in causes of death coding based on the “strengths and limitations” bullets, but didn't see it addressed again. Could this be described in the paper (methods or limitations)? Is there a concern that a systematic difference in coding of deaths could influence the results?

Many thanks for this- we have added further discussion of the choice in coding specific causes in the limitations section. We have largely followed the coding choices of Case and Deaton and subsequent follow-up papers using US data.

Results:

3. This may be beyond the scope of this manuscript, but with the inclusion of all-cause mortality, I was interested to see a bit more analysis of how the different causes of “deaths of despair” might be playing out as a proportion of mortality trends in the countries examined. A short section highlighting this might be a nice addition to the paper. It is described a bit in the Discussion section, so a short inclusion in results would tie this together.

This is a great idea, many thanks to the reviewer. While we agree that a full decomposition is beyond the scope of this paper, we have added Figures S3a-c to the supplementary materials illustrating the proportion of all-cause mortality comprised by each despair-related cause across country, time, and age. We have also calculated some illustrative examples of the proportion of change in all-cause mortality made up by drug-deaths for example in younger US age categories and have added this to the results section.

Reviewer: 3

Dr. David Walsh, Glasgow Centre for Population Health

Comments to the Author:

Thanks for the opportunity to review this study of trends in deaths from suicide, alcohol-related causes and drug-related causes in the USA, Canada and the four UK nations. The analyses seem detailed and comprehensive, reflecting a lot of work, and the paper is therefore

of interest.

My main query relates to the choice of countries which isn't really justified to any great degree – other than saying they have shared cultural and linguistic characteristics. On that basis, why not Australia and/or New Zealand? And of course there is a strong argument that, culturally, the USA is quite a different entity and the UK is more similar to other European countries (despite Brexit!). Was it a data-driven choice? It seems really important to clarify and justify the selection of countries and I'm not sure the paper, in its current form, really does that?

Thanks for this comment. We agree that it is important to clarify and justify our country selection. The selection was motivated by the fact that the UK has seen recent stalls in life expectancy, raising the question of whether it might be following in the footsteps of the US for whom these trends were visible earlier. Trends in mid-life mortality in the UK nations had not yet been described. Canada was included based on its close geographic proximity to the US, making it potentially susceptible to spill over of drug supply and other socio-political dynamics. We have now included additional justification for the country selection in the introduction.

The ICD code based definitions are also important. There has been a lot of work done in recent years in producing agreed, comparable, definitions of these causes of death. The authors' definitions seem different to others I have seen. Importantly, for suicide I don't think they include deaths from undetermined intent and for international comparisons, I think they should. More broadly, how do these definitions compare with studies employing different definitions – what (if any) are the implications?

We agree on the importance of ICD coding choices. Our coding scheme generally follows the work of Case and Deaton (2015), and subsequent refinements to this by Masters et al. 2017, with some minor differences. Our definition of suicide matches that used in Masters (U03, X60-84, Y87) and both Masters and Case & Deaton excluded deaths from undetermined intent (except where these were due to alcohol or drug poisoning, in which case they were included in those categories). Our definition of alcohol-related matches Case & Deaton except we also include alcohol poisoning (which Case & Deaton and Masters include with drug poisoning) and F10 – deaths related to alcohol dependence. Our definitions of drug-related deaths match Case & Deaton/Masters except we also include F11-16 and F18-19, deaths related to drug dependence. Given the overlapping nature of these causes of death, there will be limitations to any set of definitions. However, the conditions where we have not followed Case & Deaton/Masters are generally not major causes of death and alternative variations on our set of definitions is unlikely to change our conclusions. We have added further discussion of the coding issue in the strengths and limitations section.

There is a considerable body of evidence (international and UK based) which have attributed changes in mortality rates since the early 2010s to austerity measures which (in the UK especially) have impacted on the poorest and most vulnerable. This includes various studies of the association between austerity in England & Wales (and also Scotland) and increased drug-related deaths. It seems a huge omission for this not to be alluded to in the Discussion (especially as the impacts have been observed for around half the period you are looking at).

Many thanks. We have now added to the discussion additional context and citations on the potential links between austerity and rising mortality in the UK.

The Discussion needs amended: there is no strengths & limitations section, no summary of main findings at the start, no policy implications. There are standard sub-headings that can be used for this type of paper that people would expect to see.

We have now summarized the findings at the beginning of the discussion and added a strengths and limitations section and heading to the discussion as found in other BMJ Open papers.

Some more minor points:

- **Why choose 2001 as the starting point – is that data driven? Longer terms trends back to the 1980s and 1990s highlight a lot of important issues.**

Yes, our decision was based on the switch from ICD-9 to ICD-10 codes, which occurred around this time for the countries in our sample. For consistency, we chose to include only years where ICD-10 codes are available. Additionally, while we agree that longer time frames are important, we chose to focus on comparing countries during the more recent period of rising mid-life mortality since the early 2000s in the US originally identified by Case and Deaton (2015).

- **You analyse by quite small age groups – that’s fine, but an overall ‘premature mortality’ (<65 years) might also be important/helpful to look at**

We prefer the finer grained age-groups to examine evidence of period or cohort effects, but we have added results for the full 35-64 year-old age group to the supplemental materials (Figures S4-S5).

- **Given one of your conclusions is that “deaths of despair” isn’t actually a meaningful grouping (as trends are different for the three different causes included), is there an argument to keep them separate in the analyses (and not report them as a combined measure)?**

Given the large body of previous literature focusing on these causes as a group, we felt it important to show both the combined and disaggregated analysis. This helps to show that conclusions about trends could differ if you only consider them together.

- **In the abstract it would be helpful to define ‘deaths of despair’ earlier**

Many thanks; we have rewritten this.

- **Supreme-minor point: before a copy-editor picks up on it, your data sources should be fully referenced, not just as footnotes.**

Many thanks; we have added this.

- **Another supreme minor point: reference 19 is at the end of a sentence discussing Canada and so isn’t appropriate**

Many thanks; this is corrected.

VERSION 2 – REVIEW

REVIEWER	Jasilionis, D Max-Planck-Institut fur Demografische Forschung
REVIEW RETURNED	30-May-2023
GENERAL COMMENTS	Thank you for your answers. In my view, the raised issues by reviewers have been addressed and well explained. I do not have further comments.
REVIEWER	Walsh, David Glasgow Centre for Population Health

GENERAL COMMENTS

Many thanks for making changes – I think the manuscript is now stronger as a result (ie of all the changes made, not just (or necessarily) those in relation to my queries!).

As I said before, this represents a lot of work and it should be published – so I don't want to be annoying (honest!) and drag this out. However, quick, minor, changes to two issues would be helpful.

First, the ICD coding issue for suicide (whether or not it includes deaths from undetermined intent) is, I believe, important for international comparisons (even intra-UK comparisons) because of different legal and cultural contexts and sensitivities whereby coroners and other officials may be more or less likely to declare a death a suicide in particular places. There are various papers out there suggesting that not including undetermined intent results in an undercount. As another example of a wider issue, around 500 suicides in Northern Ireland were recently reviewed and 80% reclassified as accidental deaths (and in fact – highly relevant to your study – a great many of them were re-classified as drug-related deaths). I take your point that you have followed Deaton & Case's work but that doesn't mean they were necessarily completely correct! So I'm not for a minute suggesting you redo any of your analyses, but for your own benefit (covering your own back!) you might want to explicitly mention in the Discussion that by following Case & Deaton's approach, you exclude deaths of undetermined intent which some authors might regard as problematic for international comparisons. That will do!

Also, the mention of austerity in the Discussion could be clarified very slightly. It's not just England where evidence of austerity's impact has been shown – it's UK wide (as it was – and still is – a UK-wide set of policies); and it's not just about loss of services and old age mortality, it's affected all ages and also relates to social security cuts affecting the poorest and most in society. And the latter is directly relevant to your paper, because there is decent evidence that the dramatic increase in drug-related deaths in Scotland has been driven by horrific individual responses to either cuts to income (ie reduced social security) or a complete lack of income (through social security 'sanctions'). Evidence on this was included in Westminster cross-party enquiries into drug deaths in Scotland.

The published evidence for austerity affecting all ages (not just elderly) from different impacts includes: Richardson et al (JECH 2020), Alexiou et al (Lancet et al 2021), Martin et al (BMJ Open 2021) and lots more. In relation to austerity and drugs deaths more specifically, the Koltai et al (SSM 2021) analyses were Great Britain wide, not just England. Again, I'm not suggesting you do anything major here – just tweak the section you've written to reflect the evidence more accurately. It will take about two minutes....

Other than these two minor issues, well done on a huge amount of work and I look forward to seeing it published soon.

FAO editors: I don't need to see this again. Happy for this to be accepted on the basis of these very minor tweaks to the Discussion.

VERSION 2 – AUTHOR RESPONSE

We thank Reviewer 3 for their final two suggestions.

We have added to the methods a clarification that our suicide measure does not include deaths due to undetermined intent, as well as a sentence in the discussion as suggested to draw attention to this for those making international comparisons:

“We largely follow the original “despair” categorizations of Case and Deaton [5] and have previously demonstrated that the use of alternative categorizations is unlikely to substantively change our conclusions [50]. Our suicide category excludes deaths from undetermined intent, which differs from some studies [19] and should be considered when doing international comparisons.”

Regarding acknowledging the impact of austerity on mortality in the discussion, our current text does discuss the impact at younger ages on drug-related deaths, and we add one of the additional citations suggested by the reviewer:

“While many have focused on cuts in health and social care affecting older age mortality [46], some evidence has linked austerity to an increase in drug deaths affecting younger and middle ages [47, 48].”

We are less convinced of the strength of the current evidence on austerity and mortality, so our aim was to acknowledge this line of research but not highlight it as a primary mechanism.

We appreciate the reviewer’s positive comments and support for the publication of the paper.